# Estimation of Upper Limb Motor Function and Its Use in Activities of Daily Living Based on the Performance Time Required for the Cylinder Transfer Task in Patients with Post-Stroke Mild Hemiparesis: A Cross-Sectional Study

**DOI:** 10.3390/jcm14051591

**Published:** 2025-02-26

**Authors:** Daigo Sakamoto, Toyohiro Hamaguchi, Mina Yamamoto, Risa Aoki, Kenta Suzumura, Yasuhide Nakayama, Masahiro Abo

**Affiliations:** 1Department of Rehabilitation Medicine, The Jikei University School of Medicine Hospital, Tokyo 105-8471, Japan; daigo.0612@jikei.ac.jp (D.S.); mina1002ot@jikei.ac.jp (M.Y.); ot.lissa.0831@jikei.ac.jp (R.A.); suzumura.ot@jikei.ac.jp (K.S.); 2Department of Rehabilitation, Graduate School of Health Science, Saitama Prefectural University, Saitama 343-8540, Japan; 3Department of Rehabilitation Medicine, The Jikei University School of Medicine, Tokyo 105-8461, Japan; pt_nakayama@jikei.ac.jp

**Keywords:** stroke, activities of daily living, hemiparesis, motor function, occupational therapy

## Abstract

**Background/Objective:** Evaluating the upper limb function of the paretic and non-paretic sides of patients post-stroke is important for predicting the efficient use of the upper limbs in activities of daily living. Although there are evaluation methods that can quantify bilateral upper limb function, they are insufficient for understanding the motor characteristics of individual patients. In this study, we aimed to quantitatively evaluate bilateral upper limb function from the performance time of the cylinder transfer task of The Southampton Hand Assessment Procedure and to estimate the use status of the paralyzed upper limb. **Methods:** This cross-sectional study included 88 participants with hemiparesis post-stroke. Performance time in the three phases of the cylinder transfer task and the total performance time of these phases were measured. Moreover, existing upper limb function assessments were made. **Results:** The total performance time of the paralyzed side showed a significant correlation with the existing upper limb function assessments. A regression model was calculated to estimate the score of the existing upper limb function assessment from the performance time of each phase. **Conclusions:** This new evaluation method is a useful tool for monitoring the recovery of motor paralysis in patients post-stroke. It is our hope that clinicians will use these objective performance data to provide more effective rehabilitation treatment for patients recovering from stroke.

## 1. Introduction

Stroke is a leading cause of disability worldwide, and its sequela, hemiparesis, significantly limits patients’ activities of daily living (ADL) and reduces their quality of life (QOL) [1]. Recently, innovative therapies have been developed to restore upper limb function in patients with post-stroke hemiparesis [2,3]. These approaches are based on the neurophysiological basis of use-dependent plasticity, which strengthens the central nervous circuitry and restores function through repeated activation of specific neurons [4]. Based on this principle, active and continuous use of the paretic upper extremity is essential for patients to regain upper extremity function [5]. Therefore, an accurate and detailed assessment of bilateral upper extremity use in daily life is essential [6]. The degree of dysfunction of the patient’s paretic upper limb is quantified by comparing it to the non-paretic side [7]. The Fugl–Meyer Assessment (FMA) evaluates motor separation and coordination, and the Action Research Arm Test (ARAT) assesses the ability to grasp and manipulate objects [8]. Evaluating the upper limb function of the paretic and non-paretic sides is also important for predicting the efficient use potential of the bilateral upper limbs in ADL. For example, shoulder abduction and hand extension movements on the paretic side have been shown to be predictive of bimanual movement recovery in ADL [9]. Accurate assessment of disability can also identify the pattern of compensatory movements performed by the patient [10].

Several studies have used performance time as a parameter to differentiate between paretic and non-paretic sides and gauge the severity of motor impairments in patients with stroke [11,12]. For example, research has shown that increased performance time on the affected side is correlated with greater impairment, as measured by traditional clinical scales, such as the Fugl–Meyer Assessment of the Upper Extremity (FMA-UE) [13,14,15]. ARAT specifies a scoring method, whereby points are deducted if a movement takes abnormally long to complete on the paretic side compared to the non-paretic side; however, no clear-cut point exists for the movement time that would be deducted for each subtest [16]. The FMA-UE and ARAT are assessment methods with established reliability and validity, although they produce ceiling effects, limiting their utility for patients with mild hemiparesis [17,18]. The Box and Block Test is a simple assessment that measures the number of blocks carried in 1 min and focuses on the difference in results between the paretic and non-paretic sides, but does not assess the qualitative aspects of movement [19]. Consequently, the need for more nuanced, time-based evaluations that can detect fine motor impairments and their impact on daily functioning are required.

The Southampton Hand Assessment Procedure (SHAP) is recognized for its ability to assess hand function across various impairment levels, including mild hemiparesis, making it a valuable tool in clinical rehabilitation [20]. Its focus on timing provides a quantitative measure of task performance, which aligns with the demands of everyday activities, such as lifting, carrying, and manipulating objects. Therefore, clinicians can objectively assess improvements in motor function over time while minimizing subjective biases by focusing on the time taken to complete these tasks. The heavy-weight item in SHAP, which involves moving weighted objects, simulates real-world tasks that require both strength and dexterity. Previous studies have suggested that the heavy-weight item of SHAP is the easiest test item to perform with the highest success rate and that its performance time provides an estimate of the FMA-UE score and the paretic upper limb use in ADL [21]. However, in clinical practice, the challenges are to reduce assessment time and ensure sufficient treatment time, and the SHAP is an upper limb function test with a long assessment time, which makes applying it to patients difficult.

Therefore, this study aimed to examine the relationship between the performance time measured by the SHAP heavy-weight item instrument—developed to assess upper limb motor function in patients post-stroke—and other functional assessments. Specifically, it sought to determine their ability to discriminate between the paretic and non-paretic sides as well as estimate upper limb motor function and the use of the upper limb on the paretic side in ADL.

## 2. Materials and Methods

### 2.1. Study Design

This was a cross-sectional study. We hypothesized that the performance time for the cylinder transfer task in patients with post-stroke hemiparesis would relate to the results of other functional assessments, that the measured performance times would allow the determination of patients’ paretic and non-paretic sides, and that the performance time results would provide an estimate for patients’ upper limb motor function and paretic upper limb use in ADL.

### 2.2. Ethical Considerations

All patients provided written informed consent to participate in this study. This study was approved by the Ethics Committee of the Jikei University School of Medicine (approval number: 22-061-6238).

### 2.3. Setting

This study was conducted at the Jikei University Hospital. Five occupational therapists working at the hospital and engaged in rehabilitation in the field of cerebrovascular diseases clinically assessed the patients and measured motor tasks. The correlation test of the clinical evaluation was performed among these therapists, which confirms a similar application of the tests. Furthermore, the acquisition of patients’ medical information, clinical evaluation, and measurement of motor tasks began on 1 April 2023, and ended on 1 February 2024.

### 2.4. Participants

The eligibility criteria for the participants were patients with post-stroke hemiparesis who had undergone occupational therapy at The Jikei University Hospital between 1 April 2023, and 1 February 2024, those aged ≥18 years, and those able to grasp and release a cylinder on a table with the upper limb on the paretic side or hold an end-sitting position independently. In contrast, the exclusion criteria were impaired consciousness, mental or cognitive impairment (Mini-Mental State Examination score ≤ 25) [22] that could affect the patient’s ability to understand test instructions and perform tasks with a diagnosis of cognitive impairment post-stroke; patients with recurrent strokes; visual field impairment; patients with motor paralysis in the bilateral upper limbs; patients with sensory impairment in the upper limb on the non-paretic side; those with central nervous system or orthopedic diseases other than stroke; those with pain in the joints of the upper limbs or fingers during movement; those with marked limitations in the range of motion of the upper limbs; and patients with an amputated upper limb, hand, or fingers. Patients who met the study eligibility criteria and did not meet the exclusion criteria were asked to participate in the study, and those who consented were included. The minimum sample size was calculated as 58 patients using G*Power 3.1 (University of Dusseldorf, Dusseldorf, Germany), and the sample size was calculated by setting the test to logistic regression (z test). For calculating the required sample size, the odds ratio was 2.33, the percentage of incidence in the group with factors was 0.5, the percentage of incidence in the group with no factors was 0.3, α was 0.05, power (1 − β) was 0.8, and the distribution was log distributed.

### 2.5. Participant Characteristics

The participants’ characteristics and medical information, including sex, age, height, body weight, body mass index, dominant hand before the stroke, stroke type (ischemic or hemorrhagic), location of stroke occurrence (specific brain regions), post-stroke duration, and paretic side, were obtained from their medical records.

### 2.6. Measurement Instrument

The motor task was measured using an instrument developed for this study (Inter Reha Co., Ltd.; Tokyo, Japan, 2021). The instrument consisted of a cast aluminum cylinder and a stand with two holes (diameter = 35 mm, depth = 5 mm) equipped with an infrared light-emitting diode and sensor. The distance between the holes was 10 cm (Figure A1, Appendix A). The instrument used in this study was developed in accordance with existing SHAP specifications, but the time sensors were placed inside the instrument to enable measurement of the performance time of each phase, and an application was introduced to improve comfort. When the cylinder enters or leaves the hole in the stand, the infrared light is received, and a timestamp is automatically recorded in the Android application via Bluetooth. This device can be used to record the intervals from when the participants press the start button until they lift the cylinder (forward reach time), when the cylinder is transferred to another hole until it is placed (transfer time), and when the cylinder is placed until the participants press the end button after the cylinder is placed. The three performance times are recorded as follows: the interval from when the cylinder is placed to when the participants press the end switch (backward reach time) [Figure 1b]. Furthermore, the recorded performance time is stored as a CSV file in the internal memory of the Android application paired with the dynamometer after the participants’ IDs and measurement conditions are entered.

### 2.7. Experimental Procedure

The examiner activated the application program on the cell phone and connected it to the instrument via Bluetooth. Once the connection to the device was established, the cylinder, cylinder stand, and switch were placed in the specified locations. In the condition where the cylinder stand was placed in front of the participant (frontal task), the stand was placed 20 cm away from the bottom edge of the desk (Figure 2a). In the condition where it was placed on the ipsilateral side of the upper limb being measured (ipsilateral task), the stand was placed 20 cm away from the bottom edge of the desk and 30 cm away from the midline of the start and stop switches (Figure 2b). Next, the stand was placed in a vertical line with two holes, and the cylinder was placed in hole 1 at the far end of the stand. The manual switches were placed in front of the participant and aligned with the bottom edge of the table, with the start and end switches at the far and near ends, respectively. Additionally, the examiner checked the equipment to ensure it was positioned correctly before the measurement began.

The examiner had the participants sit in a chair with a backrest and adjusted the height of the desk so that the flexion angle of the participant’s elbow joint was 90°. Specifically, the starting position of the measurement was with the participant’s hands on the desk, with the palms of both hands facing down. The examiner demonstrated the exercise task to the participants, instructing them to press the start switch with the examiner’s hand first, to press the end switch after moving the cylinder from hole 1 in the back to hole 2 at the near end, and to perform the movement as quickly and error-free as possible. Participants were given a practice session and checked to ensure they understood the instructions given by the examiner. Measurement of the motor task was performed by an occupational therapist trained to understand the operation of the equipment and perform the measurement appropriately. This was performed in the same manner for all participants according to the written instructions.

### 2.8. Reaching Task

The movement task consisted of pressing the start button with the upper limb of the side to be measured, moving the cylinder 10 cm toward the participant, and pressing the end button (Figure 2). Measurements were performed in the following order: frontal task with the non-paretic upper limb, frontal task with the paretic upper limb, ipsilateral task with the non-paretic upper limb, and ipsilateral task with the paretic upper limb. The order of the measurements was fixed, and the same method was used for all participants. A 15 s rest period was observed between measurements. During the rest period, participants were allowed to stretch their muscles independently; however, the therapist was not allowed to provide any therapeutic intervention. If the participant failed to press the switch correctly or tipped the cylinder during the measurement, the motor task was remeasured. In contrast, the task was considered unperformable if the participant failed to perform the task on the third measurement. Forward reach, transfer, backward reach, and total performance times, which were calculated by adding the three performance times, were used for data analysis.

### 2.9. Clinical Evaluation

The FMA-UE was used to evaluate the participants’ motor paralysis [15]. Specifically, the severity of the participants’ motor paralysis was classified into five levels (no capacity: 0–22, poor capacity: 23–31, limited capacity: 32–47, notable capacity: 48–52, and full capacity: 53–66) using the FMA-UE score [23]. The Modified Ashworth Scale (MAS) was used to assess the muscle tone of the study participants. The MAS is used to evaluate spasticity, a symptom of abnormal muscle tone in central nervous system diseases [24]. In this study, the biceps brachii muscle of the paretic side of the participants was evaluated. The passive range of motion of the paretic shoulder forward flexion of the participants was measured using a goniometer. Additionally, the participants’ superficial sensation in the paretic arm and fingers was scored on a three-point ordinal scale as follows: normal, dull, or absent. The participants’ joint position sense in the upper limb was evaluated using the Thumb Search Test [25]. The Thumb Search Test is scored on a four-point ordinal scale for recognizing the position of the thumb in space. The Jikei Assessment Scale for Motor Impairment in Daily Living (JASMID) was used to investigate the use of the paretic upper limb in the participants’ daily activities [26]. With JASMID, patients answered a total of 20 questions about the amount of use of the paretic upper limb (0: never use, 3: sometimes used, and 5: always used) and their satisfaction with use (0: almost impossible to use, 3: feel moderate difficulty, and 5: feel no difficulty at all) on a five-point ordinal scale, and the quantity and quality scores were calculated.

### 2.10. Statistical Analysis

The clinical ratings, baseline information, and medical information obtained from the participants were subjected to descriptive statistics, and n (%) or median (25th, 75th percentile) was calculated. In particular, the mean, standard deviation, and minimum and maximum of the performance time were calculated for the two conditions of the task, separately for the paretic and non-paretic sides. To examine the association of the total performance time on the paretic side with the FMA-UE and JASMID scores, correlation analysis was performed to calculate the correlation coefficient. A scatterplot was created with the total performance time on the paretic side on the *x*-axis and the FMA-UE or JASMID score on the *y*-axis to visualize the relationship between the data. JASP version 0.16 (https://jasp-stats.org/, accessed on 1 March 2024) was used for these analyses.

Binomial logistic regression analysis was used to differentiate between the paretic and non-paretic sides based on total performance time. The dependent variable in this analysis was the paretic and non-paretic sides, the independent variable was total performance time, and the covariates were age, sex, and whether the dominant hand was the same as the paretic side [27,28]. After confirming the fit of the binomial logistic regression model, the receiver operating characteristic (ROC) curve was plotted, and the area under the curve was calculated. Youden’s index was used to determine the optimal cutoff value to discriminate between the paretic and non-paretic sides, and the sensitivity, specificity, positive predictive value, and negative predictive value were calculated. As a sub-analysis, we also compared the total performance time between the two groups of participants with ischemic stroke and those with hemorrhagic stroke to see if there was a difference in total performance time depending on the type of stroke.

A principal component analysis (PCA) was performed to reduce the dimensionality of the object performance time data collected from patients with post-stroke hemiparesis. The analysis used a “varimax” rotation method to extract significant components for predicting the FMA-UE and JASMID scores. Specifically, the variables of forward reach, transfer, and backward reach times measured on the paretic side were used to calculate PCA and were calculated for the frontal and ipsilateral tasks. The component scores derived from PCA were used in subsequent models. Generalized linear models (GLMs) were applied using the principal component scores as independent variables to estimate the FMA-UE and JASMID scores. The GLMs employed a gamma distribution with a log link function to account for the skewed distribution of the outcome variables (FMA-UE, JASMID quantity, and JASMID quality). Additionally, the model fit was evaluated using the Akaike Information Criterion (AIC), Bayesian Information Criterion, and deviance measures. For each outcome, the following models were constructed: FMA-UE as the dependent variable, JASMID quantity as the dependent variable, and JASMID quality as the dependent variable. Furthermore, jamovi version 2.2.1 (https://www.jamovi.org, accessed on 1 March 2024) was used for these analyses, and the statistical significance level was set at 5%. The performance times obtained using the measuring device were checked for missing values as a preliminary data processing step, and the data of participants with such values were deleted from the data set.

## 3. Results

### 3.1. Participants

During the study period, 277 patients with post-stroke hemiparesis received occupational therapy, and 90 who met the study criteria were invited to participate in the study. Ultimately, 88 patients who agreed to participate in the study were enrolled (Figure 3).

Table 1 presents the clinical characteristics of the participants. No participant had missing data for clinical assessment, basic information, or medical information. The sites of onset of cerebral infarction in the participants were as follows: middle cerebral artery region infarction in 15 participants, corona radiata infarction in 12, anterior cerebral artery region infarction in 6, medulla oblongata infarction in 4, pons infarction in 5, internal capsule infarction in 3, thalamus infarction in 3, putamen infarction in 3, parietal lobe infarction in 1, and cerebellar infarction in 1. Similarly, the sites of brain hemorrhage in the participants were as follows: 14 had a hemorrhage in the putamen, 9 in the thalamus, 8 subcortical, 3 in the internal capsule, and 1 in the pons. The post-onset period [median (25th, 75th percentile)] for the participants was 11 (7, 15) days for the 51 people in the acute phase (less than 1 month after onset); 49 (43, 50) days for the 5 people in the subacute phase (1 to 6 months after onset); and 52 (28, 83) months for the 32 people in the chronic phase (more than 6 months after onset).

### 3.2. Correlation of Performance Times with Functional Assessments

The performance times for the paretic and non-paretic sides obtained during the reach task are shown in Table 2. No missing values were found in the performance time data set.

Correlation analysis was performed on the total performance time of the paretic side, the total score of the FMA-UE, and the JASMID score, and Spearman’s rank correlation coefficient was calculated. A negative correlation was found between the total performance time of the paretic side and total score of the FMA-UE (Spearman’s rho = −0.568, *p* < 0.001) as well as between the JASMID quantity (Spearman’s rho = −0.578, *p* < 0.001) and quality (Spearman’s rho = −0.537, *p* < 0.001) scores. Additionally, a negative correlation was found between the total performance time of the paretic side in the ipsilateral task and the total score of the FMA-UE (Spearman’s rho = −0.534, *p* < 0.001), as well as between the JASMID quantity (Spearman’s rho = −0.552, *p* < 0.001) and quality (Spearman’s rho = −0.541, *p* < 0.001) scores. Figure 4 shows the scatterplot of the total performance time on the paretic side on the *x*-axis and the FMA-UE or JASMID score on the *y*-axis. The scatterplots presented in Figure 4 visualize the relationship between performance time and both the FMA-UE and JASMID scores, illustrating clear trends across different levels of motor impairment and functional ability. In both frontal and ipsilateral tasks, a sharp decline is observed in FMA-UE scores as total performance time increases, confirming that patients with greater impairments exhibit slower task completion times (Figure 4a,d). Similarly, a negative trend was observed between exercise time and the JASMID scores (both quantity and quality), confirming that patients with strongly limited ADL had slower performance times (Figure 4b,c,e,f). To address the potential ceiling effect observed in the FMA-UE and JASMID scores, a sensitivity analysis was conducted by excluding the 41 participants who scored full marks in either or both of these assessment methods. The results demonstrated that the correlation between performance time and these functional assessments remained significant (*p* < 0.001), and the overall trends were consistent with those observed in the main analysis.

### 3.3. Detection of the Paretic Upper Limb with Performance Time

Binomial logistic regression analysis performed to determine the cutoff value that discriminates the paretic side from the non-paretic side showed that the total performance time in frontal (Figure 5a) and same side (Figure 5b) tasks both fit the regression model (Table 3). Using ROC analysis, we calculated the cutoff values for discriminating between the paretic and non-paretic sides for the total performance time in the frontal and ipsilateral tasks that fit the regression model. The cutoff values for total performance time were calculated to be 3.09 and 3.22 s for the frontal and ipsilateral tasks, respectively (Table 4). In addition, as a sub-analysis, we used the Mann–Whitney U test to test whether there was a difference in total performance time between the cerebral infarction group and the cerebral hemorrhage group and found no significant difference between the two groups (mean difference = −0.0336, *p* = 0.883).

### 3.4. Estimation of the Upper Limb Motor Function and Use of the Paretic Side of the Upper Limb in ADLs by Performance Time

PCA was performed to reduce the dimensionality of the data for the performance time measured on the paretic side, and one principal component (components 1–3) was clearly identified for each of the performance times on the paretic side for the frontal and ipsilateral tasks (Table 5).

Bartlett’s test of sphericity and the Kaiser–Meyer–Olkin measure of sampling adequacy indicated that the dataset was appropriate for factor analysis. A significant factor load was found for the variable of the performance time on the paretic side in each component. The formulae for components 1–3 are shown below (Equations (1)–(3)).(1)$$\begin{array}{ll} Component\;1= & \left(0.87\times backward\;reach\;time\;of\;ipsilateral\;task\right)\\ & +\left(0.87\times forward\;reach\;time\;of\;ipsilateral\;task\right)\\ & +\left(0.82\times backward\;reach\;time\;of\;frontal\;task\right)\\ & +\left(0.76\times transfer\;time\;of\;ipsilateral\;task\right)\\ & +\left(0.75\times forward\;reach\;time\;of\;frontal\;task\right) \end{array}$$(2)$$\begin{array}{ll} Component\;2= & \left(0.84\times backward\;reach\;time\;of\;frontal\;task\right)\\ & +\left(0.78\times forward\;reach\;time\;of\;frontal\;task\right)\\ & +\left(0.69\times transfer\;time\;of\;frontal\;task\right) \end{array}$$(3)$$\begin{array}{ll} Component\;3= & \left(0.90\times forward\;reach\;time\;of\;ipsilateral\;task\right)\\ & +\left(0.85\times backward\;reach\;time\;of\;ipsilateral\;task\right)\\ & +\left(0.80\times transfer\;time\;of\;ipsilateral\;task\right) \end{array}$$

We applied GLMs using the principal component scores as independent variables to estimate the FMA-UE and JASMID scores, and the results showed that all components were significant predictors of the FMA-UE, JASMID quantity, and JASMID quality scores (Table 6). To evaluate the goodness of fit of the models, we compared the AICs of the three models and found that the AIC of component 1 was lower than that of the other models for FMA-UE, JASMID quantity, and JASMID quality.

Table 7 shows the results of the GLMs with component 1 as the independent variable. The estimated regression equations for the quantity and quality of the FMA-UE and JASMID scores were calculated. For FMA-UE, the negative coefficient for component 1 indicates that higher manipulation times are associated with lower FMA-UE scores (Equation (4)). For JASMID quantity, longer performance times lead to a lower score, reflecting reduced hand use in ADLs (Equation (5)). Similarly, for JASMID quality, this indicates that longer performance times are associated with a lower score, reflecting poorer quality of hand use in functional tasks (Equation (6)).(4)$$FMA-UE=60.49\times \exp\left(-0.05\times component\;1\right)$$(5)$$JASMID\;quantity=80.90\times \exp\left(-0.17\times component\;1\right)$$(6)$$JASMID\;quality=74.52\times \exp\left(-0.21\times component\;1\right)$$

## 4. Discussion

In this study, we hypothesized that the performance time obtained during the task of moving a cylinder on the paretic side by patients with post-stroke hemiparesis would be related to the results of other functional assessments, that the performance time could be used to determine the paretic and non-paretic sides, and that it would be possible to estimate the use of the paretic upper limb. This study demonstrated that total performance time during simple reaching tasks significantly correlates with the established clinical measures, such as the FMA-UE and JASMID. The principal component scores obtained from the performance time of each phase were shown to be significant predictors of FMA-UE, JASMID quantity, and JASMID quality scores, and the estimated regression equations for each score were calculated. These findings suggest that performance time can serve as a quick, reliable, and clinically relevant indicator of motor impairment and functional use of the upper limb in daily activities. The primary advantage of this tool lies in its simplicity and time efficiency. Compared to traditional assessments, which are usually time-consuming and require specialized training, this tool uses a straightforward reaching task and readily available technology to record performance time. Consequently, this allows for rapid assessment and helps therapists spend more time with their patients in rehabilitation. This is believed to be beneficial in busy clinical environments where time constraints are a major issue [29]. Furthermore, the quantitative nature of performance time eliminates subjective bias, providing clinicians with objective data to guide rehabilitation interventions. The correlation between performance time and functional use of the upper limb, as measured by JASMID, suggests that this tool can be used to monitor progress in ADLs.

The cut-off value for the total performance time to discriminate between the paretic and non-paretic sides was calculated to be 3.09 and 3.22 s for the frontal and ipsilateral tasks, respectively. The development of this performance time-based tool has important implications for clinical practice. By establishing cut-off values that distinguish between the paretic and non-paretic sides, this tool enables clinicians to quickly identify the affected limb. In patients post-stroke, it is easy to determine which upper limb is motor paralyzed with a simple screening test, but to clarify the difference in performance between the left and right sides, an objective and quantitative assessment is required. This point is emphasized in patients with mild hemiparesis, where ceiling effects often occur in existing upper limb function assessments. Recent research has examined the relationship between the order in which movement practice is performed on the paretic and non-paretic sides and the recovery of movement ability on the paretic side from the perspective of motor learning [30]. In addition, the non-paretic upper limb of patients post-stroke may have problems with motor control, depending on the severity of motor paralysis on the paretic side and the damaged hemisphere [31,32]. These studies show the importance of carefully evaluating changes in performance not only in the paretic arm, but also in the non-paretic arm when monitoring the recovery of motor paralysis in patients post-stroke. The evaluation method used in this study could be a useful tool for evaluating motor learning by observing changes in performance in the paretic and non-paretic arms. Notably, this is particularly valuable in early rehabilitation stages, where timely intervention is critical for optimizing recovery outcomes. Clinicians can use performance time as a practical metric to adjust rehabilitation programs based on objective performance data, enhancing the personalization of therapy [33].

Previous research has shown that the heavy-weight item in the SHAP movement task is a simple test item that patients are most likely to succeed with [21]. This is believed to be because the item can be grasped using a group flexion and partial extension movement without requiring a high level of voluntary control of the fingers, and it does not require a change in forearm position during transfer. In patients post-stroke, functional near-infrared spectroscopy and electroencephalography results measured simultaneously during finger flexion and extension motor tasks are used to analyze changes in cerebral cortical excitability and functional connectivity from immediately after stroke onset to 4 weeks later, and to verify the estimation of biomarkers for recovery of motor paralysis [34]. The evaluation method used in this study is expected to be used in clinical and research settings as a tool for monitoring the recovery of motor paralysis and predicting neurological prognosis from the perspective of movement speed in terms of the function of the distal part (which is used to grasp a cylinder and adjust its orientation) and the function of the proximal part (which is used to reach with the upper limb to transfer the cylinder). Patients with hemiparesis often experience difficulty improving motor coordination and reducing the time required to perform movements. Therefore, measuring the performance time required to complete a movement is important for assessing functional impairment and treatment response in patients with mild hemiparesis [35]. In this study, the performance time of each phase was measured, not just the total time. Since the elements of joint movement required for each phase are different, these measurements are useful for understanding the characteristics of the patient’s motor paralysis and performance in detail; thus, these phase measurements facilitate in monitoring the stage of recovery (Table A1, Appendix B). In summary, an assessment of the upper limb function that uses time spent on simple motor tasks as an outcome, such as the one used in this study, has various applications in clinical practice, and it is beneficial to use it in conjunction with existing upper limb function assessments that have a ceiling effect in patients with mild hemiparesis. The evaluation method of this study provides clinicians with data for planning individual rehabilitation treatment for patients.

One limitation of this study is the potential difficulty in generalizing the cut-off values for total performance time to different populations or types of patients with stroke. The cut-off values identified in this study were derived from a specific cohort of patients with post-stroke hemiparesis and may not apply to individuals with varying levels of impairment or different stroke etiologies. Depending on the different periods after onset, the amount of rehabilitation provided, and the differences in the programs, the participants may have different upper limb movement strategies and activity limitations [29]. The muscle strength and range of motion of patients with stroke are related to the movement patterns and ADL ability of the paretic upper limb [36,37]. In this study, the passive range of motion of the paretic shoulder forward flexion of the participants was measured, but no other assessments of joint range of motion or muscle strength were conducted. Sensory perception and muscle tone are related to the ability to manipulate objects, such as the measurement task in this study [38,39]. Although a few of the participants in this study had severe impairments, the effect of the presence or absence of these impairments on the performance time of the movement task in this study has not been clarified. In this study, participants who were able to maintain a sitting position independently participated, but the stability of the trunk—which is involved in posture control—is also involved in the quality of upper limb performance [40]. In the same way, upper limb reach movements become unstable when there is a coordination disorder such as ataxia [41]. In this study, these symptoms were not evaluated in detail, and the impact of these symptoms on exercise time was not verified. Additionally, variability may have existed in how patients were instructed to perform or how they performed the tasks, potentially introducing bias in the performance time measurements. While efforts were made to standardize task instructions and procedures, differences in patient understanding, motivation, or fatigue could have influenced performance, thereby affecting the accuracy of the performance time as an indicator of upper limb function. A notable limitation of this study is the potential ceiling effect observed in the FMA-UE and JASMID scores, particularly among participants with mild hemiparesis. This effect may lead to data points that deviate from the estimated regression curve, as observed in Figure 4. To address this, we conducted a sensitivity analysis by excluding participants with perfect scores. The results of this analysis showed that the relationships between performance time and functional assessments remained significant and aligned with the primary analysis. This suggests that, despite ceiling effects, the study findings are robust and meaningful for clinical application. Future studies should consider using evaluation tools with higher sensitivity for patients with mild impairments to minimize ceiling effects and improve measurement precision

Moving forward, this performance time-based tool has the potential for widespread application beyond stroke rehabilitation. Its ease of use and objective nature make it a promising candidate for assessing motor function in other neurological conditions, such as Parkinson’s disease, traumatic brain injury, and multiple sclerosis [42,43]. Moreover, integrating this tool with wearable technology or mobile applications could facilitate remote assessments, enabling continuous monitoring of motor function in home-based rehabilitation programs [44]. This would not only enhance patient engagement, but also allow clinicians to track recovery in real time, adjusting interventions as needed. Notably, recognizing that this study represents the first step in the development process is essential. Further research is needed to refine the tool, including the exploration of its utility across different stroke populations and in various clinical settings. Additionally, longitudinal studies are required to determine whether changes in performance time correspond to long-term improvements in upper limb function and ADL performance.

## 5. Conclusions

This study presents the development of a novel, time-efficient evaluation tool for assessing upper limb motor function in patients with stroke. By leveraging on performance time, this tool provides clinicians with a quick and objective method to evaluate motor impairment and functional use of the upper limb. While further validation is necessary, the tool holds promise for enhancing rehabilitation assessment and improving patient outcomes in various clinical contexts.

## Figures and Tables

**Figure 1 jcm-14-01591-f001:**
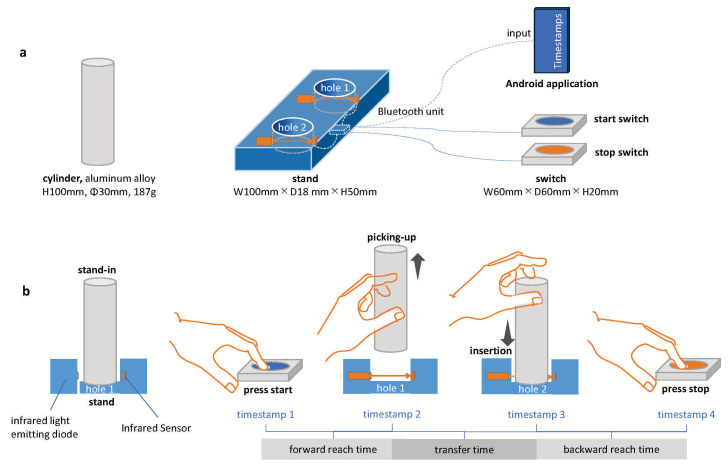
Configuration of the instrument and method of measuring motion time. (**a**) Configuration and dimensions. (**b**) Methods of measuring performance time.

**Figure 2 jcm-14-01591-f002:**
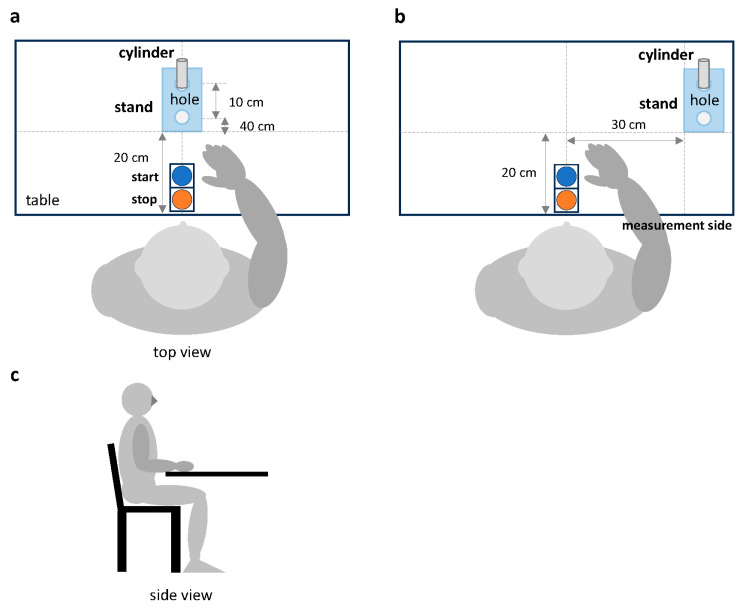
Measurement setup and limb positions of the participants. The two conditions of the motor task were (**a**) the frontal task, in which the stand was placed 20 cm away from the bottom edge of the desk; and (**b**) the ipsilateral task, in which the stand was placed 20 cm away from the bottom edge of the desk and 30 cm away from the midline of the start and stop switches. (**c**) Participants’ limb positions.

**Figure 3 jcm-14-01591-f003:**
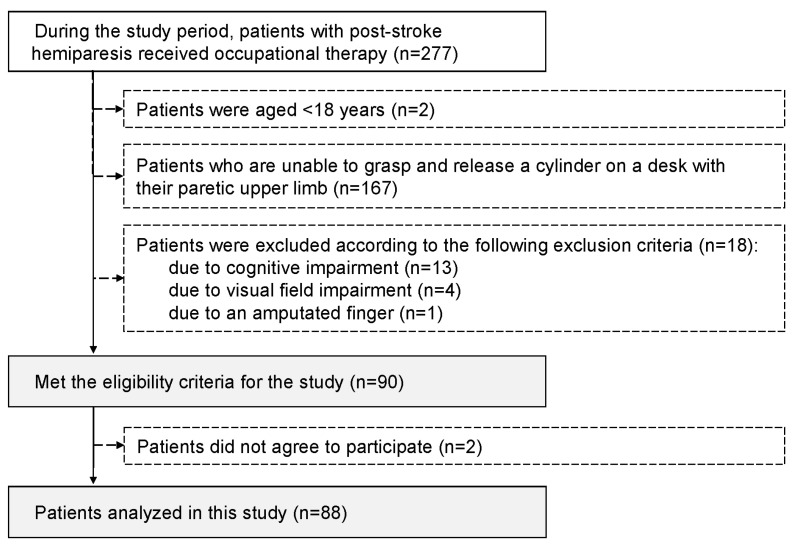
Patient selection procedure.

**Figure 4 jcm-14-01591-f004:**
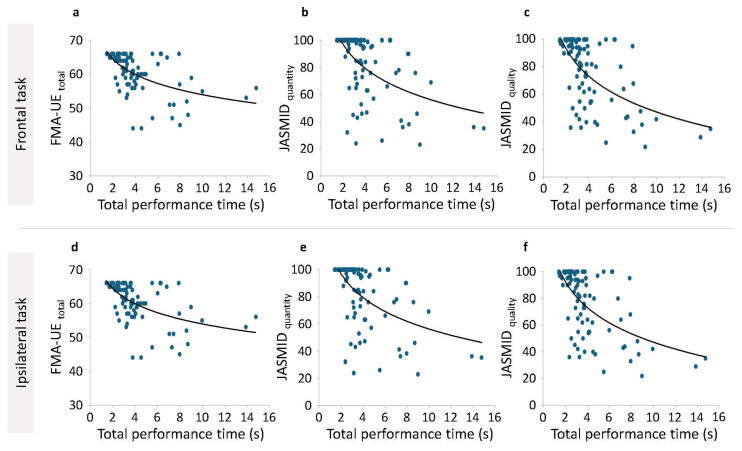
Correlation between total performance time and functional assessment scores. Scatterplots show the relationship between total performance time and FMA-UE scores (**a**,**d**), as well as between the JASMID quantity (**b**,**e**) and quality (**c**,**f**) scores in frontal (**a**–**c**) and ipsilateral (**d**–**f**) tasks. The logarithmic trend lines illustrate significant negative correlations (*p* < 0.001), indicating that longer performance times are associated with greater motor impairment and reduced upper limb use in daily activities. FMA-UE, Fugl–Meyer Assessment of the Upper Extremity; JASMID, Jikei Assessment Scale for Motor Impairment in Daily Living.

**Figure 5 jcm-14-01591-f005:**
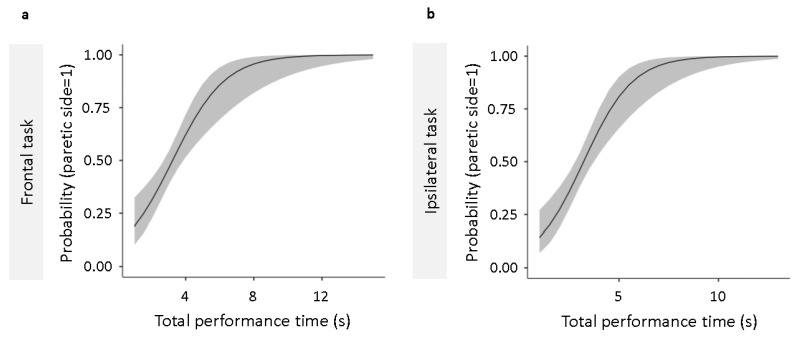
Binomial logistic regression analysis for total performance time in frontal and ipsilateral tasks. The logistic regression models predict the probability of the upper limb being on the paretic side based on total performance time. (**a**) Shows the regression curve for the frontal, and (**b**) indicates that for the ipsilateral task. The vertical axis represents the probability that the limb is paretic, with higher values indicating a greater likelihood. Both models demonstrate significant predictive accuracy (*p* < 0.001). ROC curve analysis was used to determine cut-off values for distinguishing between the paretic and non-paretic sides. FMA-UE; Fugl–Meyer Assessment of the Upper Extremity; JASMID, Jikei Assessment Scale for Motor Impairment in Daily Living; ROC, receiver operating characteristic.

**Table 1 jcm-14-01591-t001:** Clinical characteristics of the analyzed patients.

Characteristics			All (*n* = 88)
Age (years)			63 (52, 73)
Sex		Female:Male	38 (43):50 (57)
Weight (kg)			62 (52, 71)
Height (cm)			163 (156, 170)
Body mass index (kg/m^2^)			23 (21, 25)
Paretic hand		Left:Right	36 (41):52 (59)
Dominant hand		Left:Right	5 (6):83 (94)
Laterality in the paretic and dominant hand		Ipsilateral side	53 (60)
		Contralateral side	35 (40)
Diagnosis		CI	53 (60)
		ICH	35 (40)
FMA-UE		Total	62 (57, 65)
		Part A	35 (33, 36)
		Part B	10 (9, 10)
		Part C	14 (13, 14)
		Part D	5 (3, 6)
FMA-UE severity	Limited capacity (32 ≤ score ≤ 47)		5 (6)
	Notable capacity (48 ≤ score ≤ 52)		4 (4)
	Full capacity (53 ≤ score ≤ 66)		79 (90)
Modified Ashworth Scale		0	67 (76)
		1	17 (19)
		1+	4 (5)
Passive range of motion for shoulder forward flexion (°)			180 (170, 180)
Sense of touch	Finger	Normal	67 (76)
		Decline	21 (24)
	Arm	Normal	72 (82)
		Decline	16 (18)
Thumb Localizing Test		Normal	72 (82)
		Mild	10 (11)
		Moderate	5 (6)
		Severe	1 (1)
JASMID		Quantity	95 (69, 100)
		Quality	83 (55, 97)

Values are presented as n (%) or median (25th, 75th percentile). CI, cerebral infarction; FMA-UE, Fugl–Meyer Assessment of the Upper Extremity; ICH, intracranial hemorrhage; JASMID, Jikei Assessment Scale for Motor Impairment in Daily Living.

**Table 2 jcm-14-01591-t002:** Performance time during the reaching tasks.

Performance Time (s)	Paretic Side			Non-Paretic Side		
	Median	Min	Max	Median	Min	Max
Frontal task						
Total	3.31 (2.55, 4.49)	1.43	14.76	2.69 (2.13, 3.24)	1.38	5.57
Forward reach	1.09 (0.82, 1.50)	0.30	6.60	0.89 (0.72, 1.10)	0.30	2.00
Transfer	0.92 (0.72, 1,19)	0.34	3.29	0.72 (0.55, 0.98)	0.29	2.82
Backward reach	1.21 (0.93, 1,78)	0.35	10.95	0.97 (0.77, 1.16)	0.55	2.43
Ipsilateral task						
Total	3.40 (2.66, 4.72)	1.43	12.73	2.63 (2.23, 3.11)	1.54	5.91
Forward reach	1.16 (0.89, 1.80)	0.25	6.53	0.88 (0.74, 1.09)	0.25	1.96
Transfer	0.88 (0.67, 1.15)	0.29	2.97	0.77 (0.59, 0.92)	0.39	1.55
Backward reach	1.33 (1.02, 1.89)	0.32	8.06	0.98 (0.81, 1.15)	0.52	2.40

Values are presented as median (25th, 75th percentile). Max, maximum; Min, minimum.

**Table 3 jcm-14-01591-t003:** Model goodness of fit results of the binomial logistic regression analysis.

Model	Deviance	AIC	McFadden’s R^2^	Overall Model Test			Estimate	95% CI		SE	Z	*p*
				X^2^	*df*	*p*		Lower	Upper			
Total performance time of frontal task	215.48	225.48	0.12	28.50	4	<0.001	0.65	0.34	0.96	0.16	4.07	<0.001
Total performance time of ipsilateral task	208.5	218.05	0.15	35.94	4	<0.001	0.81	0.45	1.17	0.18	4.39	<0.001

AIC, Akaike’s information criterion; CI, confidence interval; SE, standard error. Binomial logistic regression was used, with the statistical significance set at 0.05 (*n* = 88).

**Table 4 jcm-14-01591-t004:** Results of the receiver operating characteristic curve analysis.

Scale	Cut off Point	Sensitivity (%)	Specificity (%)	PPV (%)	NPV (%)	Youden’s Index	AUC
Total performance time of frontal task	3.09	69.32	63.64	65.59	67.47	0.33	0.69
Total performance time of ipsilateral task	3.22	79.55	57.95	65.42	73.91	0.38	0.73

AUC, Area under the curve; NPV, Negative predictive value; PPV, Positive predictive value.

**Table 5 jcm-14-01591-t005:** Results of the principal component analysis.

Component	% of Variance	Bartlett’s Test of Sphericity			KMO Measure of Sampling Adequacy
		X^2^	df	*p*	
1	62.91	359.76	15	<0.001	0.74
2	59.99	41.34	3	<0.001	0.61
3	72.69	93.41	3	<0.001	0.67

KMO, Kaiser–Meyer–Olkin.

**Table 6 jcm-14-01591-t006:** Models calculated using the generalized linear model with component scores as independent variables.

Model	Deviance	AIC	McFadden’s R^2^	Log-Likelihood Ratio Test		
				X^2^	df	*p*
FMA-UE						
Component 1	0.63	542.64	0.28	34.89	1	<0.001
Component 2	0.64	543.93	0.26	32.88	1	<0.001
Component 3	0.64	544.57	0.26	32.29	1	<0.001
JASMID quantity						
Component 1	7.88	810.07	0.22	33.00	1	<0.001
Component 2	7.91	810.41	0.22	33.06	1	<0.001
Component 3	8.08	812.30	0.20	29.35	1	<0.001
JASMID quality						
Component 1	7.69	793.61	0.31	45.11	1	<0.001
Component 2	7.76	794.34	0.31	44.43	1	<0.001
Component 3	8.04	797.56	0.28	38.83	1	<0.001

AIC, Akaike’s information criterion; FMA-UE, Fugl–Meyer Assessment of the Upper Extremity; JASMID, Jikei Assessment Scale for Motor Impairment in Daily Living.

**Table 7 jcm-14-01591-t007:** Generalized linear model for the prediction of FMA-UE and JASMID.

Prediction	Parameter	Estimate	SE	95% CI		exp (B)	95 exp (B) CI		z	*p*
				Lower	Upper		Lower	Upper		
FMA-UE	Intercept	4.10	0.01	4.09	4.12	60.49	59.45	61.55	464.31	<0.001
	component 1	−0.05	0.01	−0.07	−0.04	0.95	0.93	0.97	−5.92	<0.001
JASMID quantity	Intercept	4.39	0.03	4.34	4.45	80.90	76.61	85.51	156.70	<0.001
	component 1	−0.17	0.03	−0.22	−0.11	0.84	0.80	0.89	−6.02	<0.001
JASMID quality	Intercept	4.31	0.03	4.25	4.37	74.52	70.34	79.04	144.83	<0.001
	component 1	−0.21	0.03	−0.27	−0.15	0.81	0.76	0.86	−7.07	<0.001

CI, confidence interval; FMA-UE, Fugl–Meyer Assessment of the Upper Extremity; JASMID, Jikei Assessment Scale for Motor Impairment in Daily Living; SE, standard error.

## Data Availability

Data presented in this study are available upon request from the corresponding author. These data are not publicly available because of privacy restrictions.

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
