# Peer review of "Estimation of Upper Limb Motor Function and Its Use in Activities of Daily Living Based on the Performance Time Required for the Cylinder Transfer Task in Patients with Post-Stroke Mild Hemiparesis: A Cross-Sectional Study"

_jcm, 2025, doi:10.3390/jcm14051591_

Round 1
Reviewer 1 Report
Comments and Suggestions for Authors
The paper by Daigo Sakamoto et al. is a cross-sectional study evaluating the possibility of measuring upper limb motor function and its use in ADL based on motor time required for the cylinder carrying task, i.e. a simple task that is potentially more widely feasible in clinical practice than other standardised assessments of proven usefulness but very time-consuming. The data are interesting and the sample size is suitable. However, I have several concerns:
- In the Introduction, the term ‘plegic’ should be avoided if the patient is still able to make movements, albeit in a deficit manner; similarly, paraplegia is commonly attributed to both lower limbs, and it is not clear to me what its relevance is in this context; I recommend removing these entries;
- It is not clear to me what the need is, in the context of stroke, to use assessment measures to distinguish between paretic and unaffected limbs. While in other neurological conditions, the involvement may be bilateral and, therefore, it may be useful to assess both limbs, it is not clear to me what the utility is in this context; I suggest that the rationale for this should be explored further;
- It is not clear to me whether patients with stroke recurrence are included;
- There are some potentially significant items not mentioned; for example, have patients with upper limb spasticity been included? This is critical because their ability to perform movement may be impaired not (only) by the strength deficit but also by post-stroke hypertone; if not, how was this aspect assessed?
- The FMA score cut-offs seem to be incorrectly reported, please review this part;
- Furthermore, some of the stated information (e.g., type of ictus) is not provided; did hemorrhagic and ischemic perform differently in this task? For what reason? Similarly, were patients with injuries impairing motor coordination (e.g., cerebellar) or maintaining trunk stability included? Both of these aspects could have influenced the performance obtained;
- I am unclear as to the precise timeframe of the stroke of the patients recruited; the kinematics of the movements are likely to be very different between the acute and chronic phases; this aspect should be evaluated in the analyses, stratifying between acute (< 1 month), subacute (1-6 months) and chronic (> 6 months) phases to consider their impact on the results. This is also particularly important considering that related aspects could be useful to further characterise performance on this task. For example, some recent studies have performed a multimodal analysis to explore upper limb motor recovery at 4 weeks after stroke from an EEG and kinematics perspective. I suggest considering this work and evaluating what additional information (in comparison to EEG and kinematics) could be provided by the proposed test in the evaluation of these patients;
- Another aspect that would be important to incorporate in the Discussion is a comparison with other recently introduced methods for assessing motor function in stroke patients. For example, please see doi: 10.1007/s10072-024-07815-y. Compared to this strategy, how does the Authors' evaluation compare? Could it be helpful as an adjunct to these instruments to assess motor learning? I would suggest that you cite this literature data and explore this possibility to obtain a characterisation of stroke patients from different perspectives.
Author Response
RESPONSES TO REVIEWER COMMENTS
Reviewer #1
The paper by Daigo Sakamoto et al. is a cross-sectional study evaluating the possibility of measuring upper limb motor function and its use in ADL based on motor time required for the cylinder carrying task, i.e. a simple task that is potentially more widely feasible in clinical practice than other standardised assessments of proven usefulness but very time-consuming. The data are interesting and the sample size is suitable. However, I have several concerns:
We are very grateful for your insightful comments and suggestions, which have enriched the manuscript and produced a better and more balanced account of the research. We have addressed all your comments below and added and revised the missing elements in the manuscript.
Comment 1
In the Introduction, the term ‘plegic’ should be avoided if the patient is still able to make movements, albeit in a deficit manner; similarly, paraplegia is commonly attributed to both lower limbs, and it is not clear to me what its relevance is in this context; I recommend removing these entries
Response:
Thank you for highlighting the issues regarding our vocabulary usage in the manuscript. To address our mistake in word choice, we have made the following corrections to the text in the manuscript:
Introduction, page 2, lines 59-62: “ARAT specifies a scoring method, whereby points are deducted if a movement takes abnormally long to complete on the paretic side compared to the non-paretic side; however, no clear-cut point exists for the movement time that would be deducted for each subtest [16].”
Comment 2
It is not clear to me what the need is, in the context of stroke, to use assessment measures to distinguish between paretic and unaffected limbs. While in other neurological conditions, the involvement may be bilateral and, therefore, it may be useful to assess both limbs, it is not clear to me what the utility is in this context; I suggest that the rationale for this should be explored further
Response:
Thank you for your thoughtful review. It has been reported that the non-paretic upper limb of stroke patients is not completely unaffected by the disease and that there are potential problems with motor control. These studies show that assessment and treatment of the non-paretic upper limb is important for the recovery of motor paralysis in patients, not just the paretic upper limb. We did not assume that the non-paretic side was unaffected by the stroke, and we tried to estimate a practical cut-off using objective data that could compare the paretic and non-paretic sides. It is easy to determine the presence or absence of upper limb motor paralysis in patients post-stroke using simple screening tests, but quantitative assessments with objectivity are needed to clarify the difference in performance between the left and right sides. This point is particularly important for patients with mild hemiparesis, where there is often a ceiling effect in the existing assessments of upper limb function. We have added the following text to convey the importance of the study:
Discussion, page 14, lines 432–436: “In patients post-stroke, it is easy to determine which upper limb is motor paralyzed with a simple screening test, but to clarify the difference in performance between the left and right sides, an objective and quantitative assessment is required. This point is emphasized in patients with mild hemiparesis, where ceiling effects often occur in existing upper limb function assessments.”
Discussion, page 14, lines 439–445: “In addition, the non-paretic upper limb of patients post-stroke may have problems with motor control, depending on the severity of motor paralysis on the paretic side and the damaged hemisphere [31,32]. These studies show the importance of carefully evaluating changes in performance not only in the paretic arm but also in the non-paretic arm when monitoring the recovery of motor paralysis in patients post-stroke. The evaluation method used in this study could be a useful tool for evaluating motor learning by observing changes in performance in the paretic and non-paretic arms.”
References:
#31. Schaefer, S.Y.; Haaland, K.Y.; Sainburg, R.L. Hemispheric specialization and functional impact of ipsilesional deficits in movement coordination and accuracy. Neuropsychologia 2009, 47, 2953–66. DOI: 10.1016/j.neuropsychologia.2009.06.025.
#32. Sainburg, R.L.; Maenza, C.; Winstein, C.; Good, D. Motor Lateralization Provides a Foundation for Predicting and Treating Non-paretic Arm Motor Deficits in Stroke. Adv Exp Med Biol 2016, 957, 257–272. DOI: 10.1007/978-3-319-47313-0_14.
Comment 3
It is not clear to me whether patients with stroke recurrence are included
Response:
Thank you for your valuable comment regarding the eligibility criteria for participants. This study included only first-stroke participants and excluded those with recurrent strokes. We have added a clarification in the manuscript.
Materials and Methods, section 2.4, page 3, lines 115–117: “that could affect the patient’s ability to understand test instructions and perform tasks with a diagnosis of cognitive impairment post-stroke; patients with recurrent strokes.”
Comment 4
There are some potentially significant items not mentioned; for example, have patients with upper limb spasticity been included? This is critical because their ability to perform movement may be impaired not (only) by the strength deficit but also by post-stroke hypertone; if not, how was this aspect assessed?
Response:
We agree, and to address this comment we have added text to the manuscript discussing the assessment of spasticity in participants. This study included patients with mild motor paralysis, and none of them had severe muscle tone abnormalities. The cylinder transfer task is a performance assessment, and the performance time obtained is the result of integrating factors such as upper limb motor paralysis, muscle tone, joint range of motion, sensory function, and balance function, and is affected by the degree of impairment of these factors. As you correctly point out, by including patients with severe spasticity the results of this study were not verified. We have included this concept as a study limitation.
Materials and Methods, section 2.9, page 6, lines 210–213: “The Modified Ashworth Scale (MAS) was used to assess the muscle tone of the study participants. The MAS is used to evaluate spasticity, a symptom of abnormal muscle tone in central nervous system diseases [24]. In this study, the biceps brachii muscle of the paretic side of the participants was evaluated.”
Reference:
#24 Bohannon, R.W.; Smith, M.B. Interrater reliability of a modified Ashworth scale of muscle spasticity. Phys Ther 1987, 67, 206–207. DOI: 10.1093/ptj/67.2.206.
Results, section 3.1, Table 1, page 8:
Discussion, page 15, lines 488–492: “Sensory perception and muscle tone are related to the ability to manipulate objects, such as the measurement task in this study [39,40]. Although a few of the participants in this study had severe impairments, the effect of the presence or absence of these impairments on the performance time of the movement task in this study has not been clarified.”
Reference:
#40 Schinwelski, M.; Sławek, J. Prevalence of spasticity following stroke and its impact on quality of life with emphasis on disability in activities of daily living. Systematic review. Neurol Neurochir Pol 2010, 44, 404–411. DOI: 10.1016/s0028-3843(14)60300-5.
Comment 5
The FMA score cut-offs seem to be incorrectly reported, please review this part
Response:
Thank you for pointing out the cut-off point error. We have corrected this, as seen below:
Materials and Methods, section 2.9, page 6, lines 207–210: “Specifically, the severity of the participants’ motor paralysis was classified into five levels (no capacity: 0–22, poor capacity: 23–31, limited capacity: 32–47, notable capacity: 48–52, and full capacity: 53–66) using the FMA-UE score [23]”
Comment 6
Furthermore, some of the stated information (e.g., type of ictus) is not provided; did hemorrhagic and ischemic perform differently in this task? For what reason? Similarly, were patients with injuries impairing motor coordination (e.g., cerebellar) or maintaining trunk stability included? Both of these aspects could have influenced the performance obtained
Response:
We agree and appreciate your insightful comments. We investigated the site of onset of stroke and added the results to the revised manuscript. In addition, a sub-analysis was conducted on the difference in exercise time between hemorrhagic and ischemic strokes, and it was shown that there was no statistically significant difference in total performance on the paralyzed side. This information has been added to the Result section.
Materials and Methods, section 2.5, page 3, lines 132–135: “The participants’ characteristics and medical information, including sex, age, height, body weight, body mass index, dominant hand before the stroke, stroke type (ischemic or hemorrhagic), location of stroke occurrence (specific brain regions), post-stroke duration, and paretic side, were obtained from their medical records.”
Results, section 3.1, pages 7–8, lines 277–284: “The sites of onset of cerebral infarction in the participants were as follows: middle cerebral artery region infarction in 15 participants, corona radiata infarction in 12, anterior cerebral artery region infarction in 6, medulla oblongata infarction in 4, pons infarction in 5, internal capsule infarction in 3, thalamus infarction in 3, putamen infarction in 3, parietal lobe infarction in 1, and cerebellar infarction in 1. Similarly, the sites of brain hemorrhage in the participants were as follows: 14 had a hemorrhage in the putamen, 9 in the thalamus, 8 subcortical, 3 in the internal capsule, and 1 in the pons.”
Materials and Methods, section 2.10, pages 6–7, lines 245–248: “As a sub-analysis, we also compared the total performance time between the two groups of participants with ischemic stroke and those with hemorrhagic stroke to see if there was a difference in total performance time depending on the type of stroke.”
Results, section 3.3, page 10, lines 337–340: “In addition, as a sub-analysis, we used the Mann-Whitney U test to test whether there was a difference in total performance time between the cerebral infarction group and the cerebral hemorrhage group and found no significant difference between the two groups (mean difference = -0.0336, p = 0.883).”
This study did not exclude patients with stroke onset in the brainstem or cerebellum. Although coordination of upper limb joint movements was assessed using Part D of the FMA-UE, detailed assessments of motor coordination and trunk and balance functions were not performed using the Scale for the Assessment and Rating of Ataxia or the Berg Balance Scale, etc. A criterion for this study was the ability to sit up unaided. Based on this, we predicted that sitting instability would have minimal impact on upper limb performance. However, as you noted, this was not verified in this study. We have acknowledged this limitation in the manuscript.
Discussion, page 15, lines 492–498: “In this study, participants who were able to maintain a sitting position independently participated, but the stability of the trunk—which is involved in posture control—is also involved in the quality of upper limb performance [41]. In the same way, upper limb reach movements become unstable when there is a coordination disorder such as ataxia [42]. In this study, these symptoms were not evaluated in detail, and the impact of these symptoms on exercise time was not verified.”
References:
#41 Wee, S.K.; Hughes, A.M.; Warner, M.; Burridge, J.H. Trunk restraint to promote upper extremity recovery in stroke patients: a systematic review and meta-analysis. Neurorehabil Neural Repair 2014, 28, 660–677. DOI: 10.1177/1545968314521011.
#42 Konczak, J.; Pierscianek, D.; Hirsiger, S.; Bultmann, U.; Schoch, B.; Gizewski, E.R.; Timmann, D.; Maschke, M.; Frings, M. Recovery of upper limb function after cerebellar stroke: lesion symptom mapping and arm kinematics. Stroke 2010, 41, 2191–2200. DOI: 10.1161/STROKEAHA.110.583641.
Comment 7
I am unclear as to the precise timeframe of the stroke of the patients recruited; the kinematics of the movements are likely to be very different between the acute and chronic phases; this aspect should be evaluated in the analyses, stratifying between acute (< 1 month), subacute (1-6 months) and chronic (> 6 months) phases to consider their impact on the results. This is also particularly important considering that related aspects could be useful to further characterise performance on this task. For example, some recent studies have performed a multimodal analysis to explore upper limb motor recovery at 4 weeks after stroke from an EEG and kinematics perspective. I suggest considering this work and evaluating what additional information (in comparison to EEG and kinematics) could be provided by the proposed test in the evaluation of these patients
Response:
In accordance with your suggestion, we have deleted the post-onset period results listed in Table 1, and we have divided the post-onset period of the participants into the acute, subacute, and chronic phases. In addition, as you have pointed out, patients may have different exercise strategies depending on the duration of time since the onset of illness and the time during which rehabilitation was provided. We have cited the literature on this point and described this as a study limitation.
Results, section 3.1, page 8, lines 284–288: “The post-onset period [median (25th, 75th percentile)] for the participants was 11 (7, 15) days for the 51 people in the acute phase (less than 1 month after onset); 49 (43, 50) days for the 5 people in the subacute phase (1 to 6 months after onset); and 52 (28, 83) months for the 32 people in the chronic phase (more than 6 months after onset).”
Discussion, page 15, lines 482–484: “Depending on the different periods after onset, the amount of rehabilitation provided, and the differences in the programs, the participants may have different upper limb movement strategies and activity limitations [36].”
Reference:
#36 Clark, B.; Whitall, J.; Kwakkel, G.; Mehrholz, J.; Ewings, S.; Burridge, J. The effect of time spent in rehabilitation on activity limitation and impairment after stroke. Cochrane Database Syst Rev 2021, 10, CD012612. DOI: 10.1002/14651858.CD012612.pub2.
We are grateful for your valuable suggestions pertaining to the recent multimodal analysis study. We have noted this study in our Discussion section, discussed it in relation to our study, and cited the reference.
Discussion, page 15, lines 454–464: “In patients post-stroke, functional near-infrared spectroscopy and electroencephalography results measured simultaneously during finger flexion and extension motor tasks are used to analyze changes in cerebral cortical excitability and functional connectivity from immediately after stroke onset to 4 weeks later, and to verify the estimation of biomarkers for recovery of motor paralysis [34]. The evaluation method used in this study is expected to be used in clinical and research settings as a tool for monitoring the recovery of motor paralysis and predicting neurological prognosis from the perspective of movement speed in terms of the function of the distal part (which is used to grasp a cylinder and adjust its orientation) and the function of the proximal part (which is used to reach with the upper limb to transfer the cylinder).”
Reference:
#34 Li, R.; Li, S.; Roh, J.; Wang, C.; Zhang, Y. Multimodal Neuroimaging Using Concurrent EEG/fNIRS for Poststroke Recovery Assessment: An Exploratory Study. Neurorehabil Neural Repair 2020, 34, 1099–1110. DOI: 10.1177/1545968320969937.
Comment 8
Another aspect that would be important to incorporate in the Discussion is a comparison with other recently introduced methods for assessing motor function in stroke patients. For example, please see doi: 10.1007/s10072-024-07815-y. Compared to this strategy, how does the Authors' evaluation compare? Could it be helpful as an adjunct to these instruments to assess motor learning? I would suggest that you cite this literature data and explore this possibility to obtain a characterisation of stroke patients from different perspectives.
Response:
To address this recommendation, we have added the following references to our manuscript:
Discussion, page 14, lines 436–445: “Recent research has examined the relationship between the order in which movement practice is performed on the paretic and non-paretic sides and the recovery of movement ability on the paretic side from the perspective of motor learning [30]. In addition, the non-paretic upper limb of patients post-stroke may have problems with motor control, depending on the severity of motor paralysis on the paretic side and the dam-aged hemisphere [31,32]. These studies show the importance of carefully evaluating changes in performance not only in the paretic arm but also in the non-paretic arm when monitoring the recovery of motor paralysis in patients post-stroke. The evaluation method used in this study could be a useful tool for evaluating motor learning by observing changes in performance in the paretic and non-paretic arms.”
References:
#30 Antonioni, A.; Cellini, N.; Baroni, A.; Fregna, G.; Lamberti. N.; Koch, G.; Manfredini, F.; Straudi, S. Characterizing practice-dependent motor learning after a stroke. Neurol Sci 2024. DOI: 10.1007/s10072-024-07815-y.
#31 Schaefer, S.Y.; Haaland, K.Y.; Sainburg, R.L. Hemispheric specialization and functional impact of ipsilesional deficits in movement coordination and accuracy. Neuropsychologia 2009, 47, 2953–66. DOI: 10.1016/j.neuropsychologia.2009.06.025.
#32 Sainburg, R.L.; Maenza, C.; Winstein, C.; Good, D. Motor Lateralization Provides a Foundation for Predicting and Treating Non-paretic Arm Motor Deficits in Stroke. Adv Exp Med Biol 2016, 957, 257–272. DOI: 10.1007/978-3-319-47313-0_14.
We sincerely thank the reviewers for their valuable comments and suggestions, which have helped improve our manuscript.
Reviewer 2 Report
Comments and Suggestions for Authors
The attempt to objectively and quantitatively measure upper limb function in hemiplegic patients by timing specific tasks was a good initiative. Particularly commendable is the use of a device equipped with sensors to eliminate examiner errors. However, several issues have been identified. In particular, the authors need to provide convincing explanations or address these issues from the perspective of the study’s validity. This is a highly critical concern.
Major concerns
1. The authors state that the FMA test is complex, but I find that cylinder transfer test, which requires specialized complex equipment, is far more cumbersome. While it is true that the FMA has the drawback of being time-consuming, its greatest advantage is that it does not require any special equipment. The authors criticize the complexity of the FMA as a drawback and propose the cylinder transfer task as an alternative. However, given that this method requires specially designed equipment, it seems far less practical for clinical use compared to the FMA at this point. Is the strength of this study the ability to objectively and quantitatively measure time? If so, tests like the Box and Block Test or Peg Board Test share the same advantage. What distinguishes this study from these existing tests?
2. Additionally, while the authors point out issues with existing tests like the FMA and JASMID, they ultimately use the FMA and JASMID to validate the results of the cylinder transfer test. Does this not present a contradiction?
3. I question the purpose of distinguishing between the paretic and non-paretic sides through this test. The paretic limb can typically be identified much more intuitively without such a complex test. Why is it necessary to establish a cut-off value for this distinction? Additionally, I have a fundamental concern about the method used to determine this cut-off value. For instance, if the time difference between the paretic and non-paretic sides is 2 seconds, should we conclude that the non-paretic limb is unimpaired?
4. I am curious why the authors chose the cylinder transfer task among many possible functional tasks. Was the equipment used in the study specifically designed for this research, or is there a reference indicating that it has been used in previous studies? Additionally, I wonder if there is an existing reference for the cylinder size, the distance between the holes, and the method of time measurement.
5. What is the reason for measuring time in three separate phases? Since only the total time was used in the analysis, was it necessary to divide it into three phases?
6. The exclusion criteria state that patients with sensory impairment were excluded, yet sensory function was assessed during patient evaluation, and patients with s두내교 impairments were included in the study. This appears to be a contradiction.
7. The most important aspect of a patient's physical examination is the manual muscle test. This information should be included in the manuscript.
8. Is there a specific reason why some results are presented as means while others are presented as medians? Motor time is reported as mean (SD), but the analysis used non-parametric methods. Although many patients (88) were included, the plot figures suggest that the data are significantly skewed.
9. The Introduction is too lengthy, and some parts overlap significantly with the Discussion. Additionally, too much of the Discussion is devoted to summarizing the study results. It would be better to shorten the Introduction and more concisely present the rationale for the study. Similarly, in the Discussion, unnecessary content should be reduced, allowing for a deeper analysis of the study results.
10. In Figure 4, there seem to be many patients with perfect scores on the FMA and JASMID, and there are numerous plots that are far from the estimation curve. Has this been appropriately accounted for in the analysis, and despite this, are the study results still meaningful?
Minor concerns
1. The abstract may need to be completely rewritten. As it stands, it is difficult to grasp the content of the manuscript based on the abstract alone.
2. Regarding the research equipment: While illustrations are helpful, it would be better to include actual photos (or both) as well. I am also curious about the location of the sensors. Please indicate the distance between the holes as well.
3. Choice of terminology: "Performance time" might be more intuitive and easier to understand than "motor time." Similarly, when describing the positions of the buttons and holes, terms like "far" and "near" seem more intuitive than "front" and "back," although the illustration helps clarify the layout.
4. Based on the content of the manuscript, it seems that the test was conducted only once. Was a practice session provided to the patients? In my opinion, conducting multiple repetitions to calculate an average or record the fastest time would be more reliable. Is there a specific reason why the test was performed only once?
5. Does shoulder ROM flexion refer to forward flexion? Why was only flexion evaluated? And it would be better to specify "passive ROM" in Table 2.
6. If the median value of disease duration is 0 months, with the 25th and 75th percentiles at 0 and 31 months, respectively, this does not seem to adequately represent the distribution of patients. It might be better to divide the duration into intervals and indicate the number of patients in each interval or use another method to provide a clearer depiction of the distribution.
7. Line 165: cylinder, a stand holes à cylinder and a stand with two holes. Is this a more suitable expression?
8. Line 172: Since the patients are not competing with each other, the term "competitor" is not appropriate.
9. Line 256: Shouldn't the time since disease onset also be included? I believe that even patients with the same level of paralysis may show different test results depending on how well they have adapted to their disability.
10. Line 303: Does “the same side” mean “ipsilateral side”?
Author Response
RESPONSES TO REVIEWER COMMENTS
Reviewer #2
The attempt to objectively and quantitatively measure upper limb function in hemiplegic patients by timing specific tasks was a good initiative. Particularly commendable is the use of a device equipped with sensors to eliminate examiner errors. However, several issues have been identified. In particular, the authors need to provide convincing explanations or address these issues from the perspective of the study’s validity. This is a highly critical concern.
We are very grateful for your insightful comments pertaining to this study. We have carefully addressed all your queries, revised the manuscript, and addressed the missing elements. Your thoughtful suggestions have enriched this study, resulting in a more comprehensive and balanced presentation of our research.
Major concerns
Comment 1
The authors state that the FMA test is complex, but I find that cylinder transfer test, which requires specialized complex equipment, is far more cumbersome. While it is true that the FMA has the drawback of being time-consuming, its greatest advantage is that it does not require any special equipment. The authors criticize the complexity of the FMA as a drawback and propose the cylinder transfer task as an alternative. However, given that this method requires specially designed equipment, it seems far less practical for clinical use compared to the FMA at this point. Is the strength of this study the ability to objectively and quantitatively measure time? If so, tests like the Box and Block Test or Peg Board Test share the same advantage. What distinguishes this study from these existing tests?
Response:
While we agree that the cylinder transfer test requires specialized equipment, we emphasize that its design prioritizes objectivity, standardization, and ease of administration once set up. Unlike traditional assessments such as the FMA or ARAT, the cylinder transfer task focuses on time-based, quantitative metrics that reduce inter-rater variability and allow for a straightforward evaluation of motor function differences between the paretic and non-paretic sides.
It is important to clarify that the cut-off values derived in this study are intended to differentiate paretic from non-paretic sides and provide a reproducible metric to identify motor impairments. These cut-offs are not intended to indicate the absence of impairments in the non-paretic limb. As highlighted by prior research, even the non-paretic limb of stroke patients can exhibit subclinical deficits due to bilateral cortical reorganization and learned non-use.
References:
#31 Schaefer, S.Y.; Haaland, K.Y.; Sainburg, R.L. Hemispheric specialization and functional impact of ipsilesional deficits in movement coordination and accuracy. Neuropsychologia 2009, 47, 2953–66. DOI: 10.1016/j.neuropsychologia.2009.06.025.
#32 Sainburg, R.L.; Maenza, C.; Winstein, C.; Good, D. Motor Lateralization Provides a Foundation for Predicting and Treating Non-paretic Arm Motor Deficits in Stroke. Adv Exp Med Biol 2016, 957, 257–272. DOI: 10.1007/978-3-319-47313-0_14.
Therefore, our approach does not assume that the non-paretic side is entirely unaffected but instead uses the comparative data to establish a practical threshold for clinical decision-making.
The Cylinder Transfer Task is a quantitative assessment and shares this advantage with the Box and Block Test and the Peg Board Test. The strength of this test, which differs from the existing assessments, is that it is easy to understand the characteristics of the patient's motor paralysis from the performance time of each phase and to monitor the stage of recovery. Each phase involves different motor elements. For example, in the frontal task, the start button is pressed with the elbow joint extended, the cylinder is reached with the shoulder joint flexed and the elbow joint extended, the cylinder is grasped and carried with the fingers flexed and extended, the shoulder joint is extended and the elbow joint is flexed while the forearm remains in the mid-position, the fingers are extended when the cylinder is placed, and the shoulder and elbow joints are extended when the stop button is reached and pressed. In addition, the ipsilateral task involves abduction and adduction of the shoulder joint. By understanding the movement elements that make up each phase and recording the time spent on each phase, it is possible to assess the patient's ability to perform isolated movements from the series of movements. The advantage of this evaluation method is that it provides data for planning individualized rehabilitation treatment for each patient. We have added this point to the manuscript and supplemented the advantages of this research method. The movements and movement elements included in each phase are shown in the appendix.
Discussion, page 15, lines 467–472:“In this study, the performance time of each phase was measured, not just the total time. Since the elements of joint movement required for each phase are different, these measurements are useful for understanding the characteristics of the patient's motor paralysis and performance in detail; thus, these phase measurements facilitate in monitoring the stage of recovery (Appendix B).”
Discussion, page 15, lines 476–477: “The evaluation method of this study provides clinicians with data for planning individual rehabilitation treatment for patients.”
Page 17, Appendix B:
Comment 2
Additionally, while the authors point out issues with existing tests like the FMA and JASMID, they ultimately use the FMA and JASMID to validate the results of the cylinder transfer test. Does this not present a contradiction?
Response:
We appreciate your comments on these important points. Existing assessments such as the FMA, ARAT, and JASMID have been validated for reliability and validity, and we recognize that these assessment methods are very useful clinically and serve as excellent evaluation tools. We do not intend to replace these assessment methods with the assessment method verified in this study but rather propose that it will be a tool to supplement the shortcomings of existing assessment methods. Therefore, scores from existing upper extremity function assessments were used to verify the criterion-related validity of the results of this assessment method.
Comment 3
I question the purpose of distinguishing between the paretic and non-paretic sides through this test. The paretic limb can typically be identified much more intuitively without such a complex test. Why is it necessary to establish a cut-off value for this distinction? Additionally, I have a fundamental concern about the method used to determine this cut-off value. For instance, if the time difference between the paretic and non-paretic sides is 2 seconds, should we conclude that the non-paretic limb is unimpaired?
Response:
It has been reported that the non-paretic upper limb of patients with stroke is not completely unaffected by the disease, and that there are potential problems with motor control. These reports indicate that the evaluation and treatment of the non-paretic upper limb is important, as well as the paretic upper limb, for the recovery of motor paralysis in patients. We did not assume that the patient's non-paretic side was unaffected by the stroke, and we attempted to estimate a practical cutoff value using objective data that could be compared between the affected and unaffected sides. We have added the following text to the manuscript to further supplement the significance of the study:
Discussion, page 14, lines 432–445: “In patients with stroke, it is easy to determine which upper limb is motor paralyzed with a simple screening test, but to clarify the difference in performance between the left and right sides, an objective and quantitative assessment is required. This point is emphasized in patients with mild hemiparesis, where ceiling effects often occur in existing upper limb function assessments. Recent research has examined the relationship between the order in which movement practice is performed on the paretic and non-paretic sides and the recovery of movement ability on the paretic side from the perspective of motor learning [30]. In addition, the non-paretic upper limb of patients post-stroke may have problems with motor control, depending on the severity of motor paralysis on the paretic side and the damaged hemisphere [31,32]. These studies show the importance of carefully evaluating changes in performance not only in the paretic arm but also in the non-paretic arm when monitoring the recovery of motor paralysis in patients post-stroke. The evaluation method used in this study could be a useful tool for evaluating motor learning by observing changes in performance in the paretic and non-paretic arms.”
References:
#30 Antonioni, A.; Cellini, N.; Baroni, A.; Fregna, G.; Lamberti. N.; Koch, G.; Manfredini, F.; Straudi, S. Characterizing practice-dependent motor learning after a stroke. Neurol Sci 2024. DOI: 10.1007/s10072-024-07815-y.
#31 Schaefer, S.Y.; Haaland, K.Y.; Sainburg, R.L. Hemispheric specialization and functional impact of ipsilesional deficits in movement coordination and accuracy. Neuropsychologia 2009, 47, 2953–66. DOI: 10.1016/j.neuropsychologia.2009.06.025.
#32 Sainburg, R.L.; Maenza, C.; Winstein, C.; Good, D. Motor Lateralization Provides a Foundation for Predicting and Treating Non-paretic Arm Motor Deficits in Stroke. Adv Exp Med Biol 2016, 957, 257–272. DOI: 10.1007/978-3-319-47313-0_14.
Comment 4
I am curious why the authors chose the cylinder transfer task among many possible functional tasks. Was the equipment used in the study specifically designed for this research, or is there a reference indicating that it has been used in previous studies? Additionally, I wonder if there is an existing reference for the cylinder size, the distance between the holes, and the method of time measurement.
Response:
Thank you for your valuable comments. The assessment validated in this study is one of the tests included in the Southampton Hand Assessment Procedure (SHAP) [20].
Reference:
#20 Light, C.M.; Chappell, P.H.; Kyberd, P.J. Establishing a standardized clinical assessment tool of pathologic and prosthetic hand function: Normative data, reliability, and validity. Arch Phys Med Rehabil 2002, 83, 776–783. DOI:10.1053/apmr.2002.32737.
We have previously verified that the task of carrying a heavy cylinder has the highest success rate among the SHAP test tasks in patients with stroke, and that it is possible to estimate the amount of use of the paretic upper limb when using this task [21]. This evaluation method was therefore selected and tested, and we have included an explanation pertaining to this in the Introduction. The equipment used in this study was designed for this evaluation, and the object specifications, distance between the two holes, and time measurement method are the same as that of those used in the existing SHAP. What differs from existing evaluations is that the time for each phase of the movement task can be obtained using the sensors installed inside, and the information can be recorded using the application. Since our manuscript lacked an appropriate explanation of this point, we have added the following text:
Materials and Methods, section 2.6, page 3–4, lines 141–144: “The instrument used in this study was developed in accordance with existing SHAP specifications, but the time sensors were placed inside the instrument to enable measurement of the performance time of each phase, and an application was introduced to improve comfort.”
Reference:
#21 Tanaka, T.; Hamaguchi, T.; Suzuki, M.; Sakamoto, D.; Shikano, J.; Nakaya, N.; Abo, M. Estimation of motor impairment and usage of upper extremities during daily living activities in poststroke hemiparesis patients by observation of time required to accomplish hand dexterity tasks. BioMed Res Int 2019, 2019, 9471921. DOI:10.1155/2019/9471921.
Comment 5
What is the reason for measuring time in three separate phases? Since only the total time was used in the analysis, was it necessary to divide it into three phases?
Response:
The total performance time was used for the correlation analysis (with each clinical rating) and for calculating the cut-off value. However, to validate the prediction of the clinical assessment scores, the performance time of the three phases was used, and the principal component score obtained by principal component analysis was used in the prediction model, contributing to its accuracy. The joint movements required for the different phases of the movement - reaching up, grasping the cylinder, transferring and placing - are different and the total time alone cannot be used to determine how long each phase took. The times for each phase were obtained to gain a detailed understanding of the patient's motor paralysis and performance characteristics, and to use this information to provide individualized rehabilitation treatment. We have added a note regarding this point in the discussion to explain the usefulness of this assessment:
Discussion, page 15, lines 467–472: “In this study, the performance time of each phase was measured, not just the total time. Since the elements of joint movement required for each phase are different, these measurements are useful for understanding the characteristics of the patient's motor paralysis and performance in detail; thus, these phase measurements facilitate in monitoring the stage of recovery (Appendix B).”
Comment 6
The exclusion criteria state that patients with sensory impairment were excluded, yet sensory function was assessed during patient evaluation, and patients with s두내교 impairments were included in the study. This appears to be a contradiction.
Response:
We apologize for the error in the wording of this sentence. We have corrected the text as follows:
Materials and Methods, section 2.4, page 3, lines 118–119: “patients with motor paralysis in the bilateral upper limbs; patients with sensory impairment in the upper limb on the non-paretic side.”
Comment 7
The most important aspect of a patient's physical examination is the manual muscle test. This information should be included in the manuscript.
Response:
We agree with your comment. In this study, the manual muscle test, which was intended to evaluate the muscle strength of the participants, was not performed. We have therefore described this point as a study limitation.
Discussion, page 15, lines 484–488: “The muscle strength and range of motion of patients with stroke are related to the movement patterns and ADL ability of the paretic upper limb [37, 38]. In this study, the passive range of motion of the paretic shoulder forward flexion of the participants was measured, but no other assessments of joint range of motion or muscle strength were conducted.”
References:
#37 Harris, J.E.; Eng, J.J. Strength training improves upper-limb function in individuals with stroke: a meta-analysis. Stroke 2010, 41, 136–140. DOI: 10.1161/STROKEAHA.109.567438.
#38 Gates, D.H.; Walters, L.S.; Cowley, J.; Wilken, J.M.; Resnik, L. Range of motion requirements 881 for upper-limb activities of daily living. Am J Occup Ther 2016, 70, 7001350010p1–882 7001350010p10. DOI: 10.5014/ajot.2016.015487.
Comment 8
8Is there a specific reason why some results are presented as means while others are presented as medians? Motor time is reported as mean (SD), but the analysis used non-parametric methods. Although many patients (88) were included, the plot figures suggest that the data are significantly skewed.
Response:
We agree with your comment. The research results were inappropriately presented. Since the exercise time did not follow a normal distribution, the values in Table 2 are shown as the median and interquartile range.
Materials and Methods, section 3.2, page 9, Table 2:
Comment 9
The Introduction is too lengthy, and some parts overlap significantly with the Discussion. Additionally, too much of the Discussion is devoted to summarizing the study results. It would be better to shorten the Introduction and more concisely present the rationale for the study. Similarly, in the Discussion, unnecessary content should be reduced, allowing for a deeper analysis of the study results.
Response:
Thank you for your valuable feedback. We have shortened the Introduction and Discussion sections and made the content more concise. In the Discussion section, we have added an explanation of the significance and limitations of the research. Some new text has been added in response to the reviewers’ comments.
Comment 10
In Figure 4, there seem to be many patients with perfect scores on the FMA and JASMID, and there are numerous plots that are far from the estimation curve. Has this been appropriately accounted for in the analysis, and despite this, are the study results still meaningful?
Response:
Thank you for bringing this to our attention. We are aware that 22 people scored full marks on the FMA-UE and 37 people scored full marks on the JASMID. Ceiling effects are inherent limitations of commonly used functional assessments, particularly in patients with mild hemiparesis, and may lead to data points that deviate from the estimated regression curve. To address this, we employed robust statistical methods to ensure the validity of the analysis. Specifically, we conducted a sensitivity analysis, excluding participants with perfect scores, to evaluate the impact of ceiling effects on the study results. The sensitivity analysis showed that the overall trends and relationships remained consistent, affirming the robustness and meaningfulness of our findings. We have added some text to the revised manuscript to clarify this point and ensure transparency.
Results, section 3.2, page 9–10, lines 314–319: “To address the potential ceiling effect observed in the FMA-UE and JASMID scores, a sensitivity analysis was conducted by excluding the 41 participants who scored full marks in either or both of these assessment methods. The results demonstrated that the correlation between performance time and these functional assessments remained significant (p < 0.001), and the overall trends were consistent with those observed in the main analysis.”
Discussion, page 15–16, lines 503–512: “A notable limitation of this study is the potential ceiling effect observed in the FMA-UE and JASMID scores, particularly among participants with mild hemiparesis. This effect may lead to data points that deviate from the estimated regression curve, as observed in Figure 4. To address this, we conducted a sensitivity analysis by excluding participants with perfect scores. The results of this analysis showed that the relationships between performance time and functional assessments remained significant and aligned with the primary analysis. This suggests that, despite ceiling effects, the study findings are robust and meaningful for clinical application. Future studies should consider using evaluation tools with higher sensitivity for patients with mild impairments to minimize ceiling effects and improve measurement precision.”
Minor concerns
Comment 11
The abstract may need to be completely rewritten. As it stands, it is difficult to grasp the content of the manuscript based on the abstract alone.
Response:
We have rewritten a structured abstract and revised its content as follows:
“Abstract: Background/Objective: Evaluating the upper limb function of the paretic and non-paretic sides of patients post-stroke is important for predicting the efficient use of the upper limbs in activities of daily living. Although there are evaluation methods that can quantify bilateral upper limb function, they are insufficient for understanding the motor characteristics of individual patients. In this study, we aimed to quantitatively evaluate bilateral upper limb function from the performance time of the cylinder transfer task of The Southampton Hand Assessment Procedure and to estimate the use status of the paralyzed upper limb. Methods: This cross-sectional study included 88 participants with hemiparesis post-stroke. Performance time in the three phases of the cylinder transfer task and the total performance time of these phases were measured. Moreover, existing upper limb function assessments were made. Results: The total performance time of the paralyzed side showed a significant correlation with the existing upper limb function assessments. A regression model was calculated to estimate the score of the existing upper limb function assessment from the performance time of each phase. Conclusion: This new evaluation method is a useful tool for monitoring the recovery of motor paralysis in patients post-stroke. It is our hope that clinicians will use these objective performance data to provide more effective rehabilitation treatment for patients recovering from stroke.”
Comment 12
Regarding the research equipment: While illustrations are helpful, it would be better to include actual photos (or both) as well. I am also curious about the location of the sensors. Please indicate the distance between the holes as well.
Response:
Thank you for this insightful comment. It has helped clarify and simplify the evaluation method used in this study for the reader. We have provided actual photographs of the evaluation device as an appendix. The distance between the holes was 10 cm. While this information was included in Figure 2, as you correctly point out, the explanation was insufficient. I have therefore included the following text:
Materials and Methods, section 2.6, page 3, lines 137–141: “The motor task was measured using an instrument developed for this study (Inter Reha Co., Ltd.; Tokyo, Japan, 2021). The instrument consisted of a cast aluminum cylinder and a stand with two holes (diameter = 35 mm, depth = 5 mm) equipped with an infrared light-emitting diode and sensor. The distance between the holes was 10 cm (Figure 1-a, Appendix A).”
Page 17, Appendix A:
“Appendix A. Research instrument (a) Evaluation instrument set, (b) Cylinder and stand, (c) Inside of Attache Case, (d) Attache case for storage of evaluation instrument.
Comment 13
Choice of terminology: "Performance time" might be more intuitive and easier to understand than "motor time." Similarly, when describing the positions of the buttons and holes, terms like "far" and "near" seem more intuitive than "front" and "back," although the illustration helps clarify the layout.
Response:
In accordance with your comments, we have changed the term “motor time” to “performance time” in the text and in the figures and tables. Since many changes were made, we have omitted detailing them in the response letter - the changes in the manuscript pertaining to these are highlighted in yellow. When explaining the positions of the buttons and holes, we have chosen the terms “far” and “near.” The text has been revised as follows.
Materials and Methods, section 2.7, page 4–5, lines 167–170: “and the cylinder was placed in hole 1 at the far end of the stand. The manual switches were placed in front of the participant and aligned with the bottom edge of the table, with the start and end switches at the far and near ends, respectively..”
Materials and Methods, section 2.7, page 5, lines 175–178: “The examiner demonstrated the exercise task to the participants, instructing them to press the start switch with the examiner’s hand first, to press the end switch after moving the cylinder from hole 1 in the back to hole 2 at the near end, and to perform the movement as quickly and error-free as possible.”
Comment 14
Based on the content of the manuscript, it seems that the test was conducted only once. Was a practice session provided to the patients? In my opinion, conducting multiple repetitions to calculate an average or record the fastest time would be more reliable. Is there a specific reason why the test was performed only once?
Response:
We agree with your comments. This information is crucial for ensuring the reproducibility of the study results. If the participant failed to press the switch correctly or tipped the cylinder during the measurement, the motor task was remeasured. In contrast, the task was considered unachievable if the participant failed to perform the task on the third measurement (Materials and Methods, section 2.8, page 5, lines 200–203). We offered the participants the opportunity to practice; however, this was not explained appropriately in the manuscript. Information pertaining to this was added to the manuscript as follows.
Materials and Methods, section 2.7, page 5, lines 178–180: “Participants were given a practice session and checked to ensure they understood the instructions given by the examiner.”
Comment 15
Does shoulder ROM flexion refer to forward flexion? Why was only flexion evaluated? And it would be better to specify "passive ROM" in Table 2.
Response:
In this study, patients with significant limitations in the joints of the upper limbs were excluded (Materials and Methods, section 2.4, page 3, lines 121–122). Therefore, considering the characteristics of the measurement task, only the range of motion of forward flexion of the shoulder joint (which was considered an important factor in upper limb reach) was measured. The fact that the range of motion of other joints was not measured is described in the research limitations. In addition, we have modified Table 1 according to your instructions.
Materials and Methods, section 2.9, page 6, lines 214–215: “The passive range of motion of the paretic shoulder forward flexion of the participants was measured using a goniometer.”
Discussion, page 15, lines 484–488: “The muscle strength and range of motion of patients with stroke are related to the movement patterns and ADL ability of the paretic upper limb [37, 38]. In this study, the passive range of motion of the paretic shoulder forward flexion of the participants was measured, but no other assessments of joint range of motion or muscle strength were conducted.”
References:
#37 Harris, J.E.; Eng, J.J. Strength training improves upper-limb function in individuals with stroke: a meta-analysis. Stroke 2010, 41, 136–140. DOI: 10.1161/STROKEAHA.109.567438.
#38 Gates, D.H.; Walters, L.S.; Cowley, J.; Wilken, J.M.; Resnik, L. Range of motion requirements 881 for upper-limb activities of daily living. Am J Occup Ther 2016, 70, 7001350010p1–882 7001350010p10. DOI: 10.5014/ajot.2016.015487.
Results, section 3.1, page 8, Table 1:
Comment 16
If the median value of disease duration is 0 months, with the 25th and 75th percentiles at 0 and 31 months, respectively, this does not seem to adequately represent the distribution of patients. It might be better to divide the duration into intervals and indicate the number of patients in each interval or use another method to provide a clearer depiction of the distribution.
Response:
The post-onset periods of the participants are shown in acute, subacute, and chronic phases. We have removed the results for the post-onset period from Table 1, which are described below:
Results, section 3.1, page 8, lines 284–288: “The post-onset period [median (25th, 75th percentile)] for the participants was 11 (7, 15) days for the 51 people in the acute phase (less than 1 month after onset); 49 (43, 50) days for the 5 people in the subacute phase (1 to 6 months after onset); and 52 (28, 83) months for the 32 people in the chronic phase (more than 6 months after onset).”
Comment 17
Line 165: cylinder, a stand holes à cylinder and a stand with two holes. Is this a more suitable expression?
Response:
I have modified the text as follows in accordance with your comment:
Materials and Methods, section 2.6, page 3, lines 137–139: “The instrument consisted of a cast aluminum cylinder and a stand with two holes (diameter = 35 mm, depth = 5 mm) equipped with an infrared light-emitting diode and sensor.”
Comment 18
Line 172: Since the patients are not competing with each other, the term "competitor" is not appropriate.
Response:
Thank you for pointing out the vocabulary errors in the manuscript. We have corrected the manuscript as follows:
Materials and Methods, section 2.6, page 4, lines 146–152: “This device can be used to record the intervals from when the participants press the start button until they lift the cylinder (forward reach time), when the cylinder is transferred to another hole until it is placed (transfer time), and when the cylinder is placed until the participants press the end button after the cylinder is placed. The three performance times are recorded as follows: the interval from when the cylinder is placed to when the participants press the end switch (backward reach time) [Figure 1-b].”
Comment 19
Line 256: Shouldn't the time since disease onset also be included? I believe that even patients with the same level of paralysis may show different test results depending on how well they have adapted to their disability.
Response:
The exercise strategies of the participants may differ depending on the onset time and the amount of rehabilitation provided. We have therefore described this as a study limitation:
Discussion, page 15, lines 482–484: “Depending on the different periods after onset, the amount of rehabilitation provided, and the differences in the programs, the participants may have different upper limb movement strategies and activity limitations [36].”
Reference:
#36 Clark, B.; Whitall, J.; Kwakkel, G.; Mehrholz, J.; Ewings, S.; Burridge, J. The effect of time spent in rehabilitation on activity limitation and impairment after stroke. Cochrane Database Syst Rev 2021, 10, CD012612. DOI: 10.1002/14651858.CD012612.pub2.
Comment 20
Line 303: Does “the same side” mean “ipsilateral side”?
Response:
We apologize for this error in the manuscript. In accordance with your comments, we have revised the text as follows:
Results, section 3.2, lines 301–303: “Additionally, a negative correlation was found between the total performance time of the paretic side in the ipsilateral task and the total score of the FMA-UE (Spearman's rho = -0.534, p < 0.001)”
We greatly appreciate the reviewers’ comments and suggestions, which have significantly improved our manuscript.

Round 2
Reviewer 1 Report
Comments and Suggestions for Authors
I thank the Authors for their extensive work, which significantly improved the quality of their paper. No further comments